# Improving the Sparse Structure Learning of Spiking Neural Networks from the View of Compression Efficiency

**Jiangrong Shen** †‡§ß    **Qi Xu** ¶*    **Gang Pan** §    **Badong Chen** ‡ß

†Faculty of Electronic and Information Engineering, Xi'an Jiaotong University
‡Institute of Artificial Intelligence and Robotics, Xi'an Jiaotong University
§State Key Lab of Brain-Machine Intelligence, Zhejiang University
¶School of Computer Science, Dalian University of Technology
ßNational Key Lab of Human-Machine Hybrid Augmented Intelligence, Xi'an Jiaotong University

## Abstract

The human brain utilizes spikes for information transmission and dynamically reorganizes its network structure to boost energy efficiency and cognitive capabilities throughout its lifespan. Drawing inspiration from this spike-based computation, Spiking Neural Networks (SNNs) have been developed to construct event-driven models that emulate this efficiency. Despite these advances, deep SNNs continue to suffer from over-parameterization during training and inference, a stark contrast to the brain's ability to self-organize. Furthermore, existing sparse SNNs are challenged by maintaining optimal pruning levels due to a static pruning ratio, resulting in either under or over-pruning. In this paper, we propose a novel two-stage dynamic structure learning approach for deep SNNs, aimed at maintaining effective sparse training from scratch while optimizing compression efficiency. The first stage evaluates the compressibility of existing sparse subnetworks within SNNs using the PQ index, which facilitates an adaptive determination of the rewiring ratio for synaptic connections based on data compression insights. In the second stage, this rewiring ratio critically informs the dynamic synaptic connection rewiring process, including both pruning and regrowth. This approach significantly improves the exploration of sparse structures training in deep SNNs, adapting sparsity dynamically from the point view of compression efficiency. Our experiments demonstrate that this sparse training approach not only aligns with the performance of current deep SNNs models but also significantly improves the efficiency of compressing sparse SNNs. Crucially, it preserves the advantages of initiating training with sparse models and offers a promising solution for implementing edge AI on neuromorphic hardware.

## 1 Introduction

Spiking Neural Networks (SNNs) have garnered increasing attention due to their event-driven properties, high spatiotemporal dynamics, and structural and learning plasticity that mimic biological neural processing (Maass, 1997; Subbulakshmi Radhakrishnan et al., 2021; Fang et al., 2023; Hu et al., 2023). Unlike traditional artificial neural networks that rely on continuous signal computation, SNNs process information using discrete events, aligning more closely with the energy-efficient mechanisms observed in human neural activity. The post-synaptic neurons in SNNs receive spike trains from pre-synaptic neurons and emit output spikes upon crossing a firing threshold (Stanojevic et al., 2024; Zhou et al., 2023; Hu et al., 2021). Consequently, bio-inspired SNNs offer significant advantages in energy efficiency, making them especially suitable for neuromorphic computing in Edge AI such as event-based vision (Liu et al., 2024b;c; 2025), where energy constraints are

---

*Accepted by ICLR 2025. The corresponding author: xuqi@dlut.edu.cn.

paramount (Imam & Cleland, 2020; Pei et al., 2019; Deng et al., 2021a). Despite these inherent advantages, the deployment of increasingly deep SNNs introduces substantial challenges, particularly over-parameterization during training and inference, leading to excessive computational overhead and memory usage. This misalignment with the resource-efficient requirements of edge devices calls for innovative solutions.

Current research on the sparse structure learning of deep SNNs aims to address the over-parameterization issue. These methodologies are predominantly categorized by their computational cost throughout the whole training process. The first one is the gradually structural sparsification approach with the non-sparse network as the initial status. For instance, the gradient reparameterization (Chen et al., 2021; 2022) approach implements the gradually efficient sparsification for deep SNNs with learnable pruning speed by redefining the weight parameters and threshold growing function. Alternatively, the second category called fully sparsification methods initiate with sparse SNNs to maintain connection sparsity throughout training, exemplified by sparse evolutionary rewiring (Shen et al., 2023) and the lottery-ticket hypothesis (Kim et al., 2022). We advocate for the latter due to its compatibility with hardware constraints like on-chip training. Considering the intrinsic connection between network sparsity and compressibility, the PQ index has been proven to satisfy the six properties of an ideal sparsity measure (Hurley & Rickard, 2009), and employed as the indicator of vector sparsity in the traditional artificial neural networks in (Diao et al., 2023). However, that sparsity measure analysis ignores the unique spatial and temporal dynamics in SNNs. Therfore, most existing fully sparse training methods for SNNs employ static pruning ratios or predetermined sparsity levels, ignoring the analysis of spatiotemporal dynamics of SNNs and lacking the necessary flexibility like self-reorganizing in human brain and often leading to under or over-pruning.

Observing the brain's flexible organization of large-scale functional networks, which adapt through environmental interactions, offers a clue towards solving deep SNNs' over-parameterization. During brain development, synaptic connections undergo structural plasticity, forming new synapses and eliminating existing ones (De Vivo et al., 2017; Barnes & Finnerty, 2010; Bennett et al., 2018). This rewiring process forms flexible network structure and promotes synaptic sparsity contributing to the brain's low power consumption. Therefore, emulating the brain's structural synaptic plasticity through a dynamic structure learning approach could be key to developing more flexible deep SNN models. Meanwhile, the density of synaptic connections in the brain optimizes throughout development, although precise control mechanisms remain unclear. Thus, we explore optimizing the pruning ratio from a neural network compression perspective as explored in machine learning, where data compression theory helps quantify the compressibility of a sub-network during each connection updating iteration, thereby avoiding under or over-pruning (Neill, 2020). By combining with the biologically plausible rewiring mechanism in human brain and neural network compression theory in machine learning, we attempt to give a solution about the sparse training method from scratch for deep SNNs with adaptive and suitable pruning ratio setting.

In light of these above insights, this paper proposes a novel two-stage sparse structure learning method from scratch for SNNs, utilizing the PQ index to measure appropriate compressibility. This method not only maintains sparse training throughout the learning process but also mitigates the issues of under-pruning and over-pruning in sparse SNNs. Our contributions are as follows:

- We introduce a pioneering two-stage dynamic structure learning framework for deep SNNs that utilizes the PQ index to gauge and dynamically adjust structure of sparse subnetworks according to compressibility. This novel approach tailors the rewiring ratio throughout the training process, providing a fine-tuned, adaptive mechanism that enhances the foundational training dynamics of deep sparse SNNs.

- Our methodology extends traditional sparse training approaches for SNNs by implementing a continuous, iterative learning process across two stages. In the first stage, the PQ index informs the adjustment of synaptic connection rewiring ratios. In the second stage, these ratios guide a dynamic rewiring strategy that includes both the pruning and regrowth of connections. Thus the methodology optimizes the SNNs' structural efficiency and operational effectiveness far beyond conventional static pruning techniques.

- Through extensive empirical testing, our method not only achieves competitive performance relative to existing state-of-the-art models but also significantly enhances the efficiency of sparse training from scratch implementations for deep SNNs. This rigorous validation demonstrates our approach's ability to maintain essential SNNs' network func-

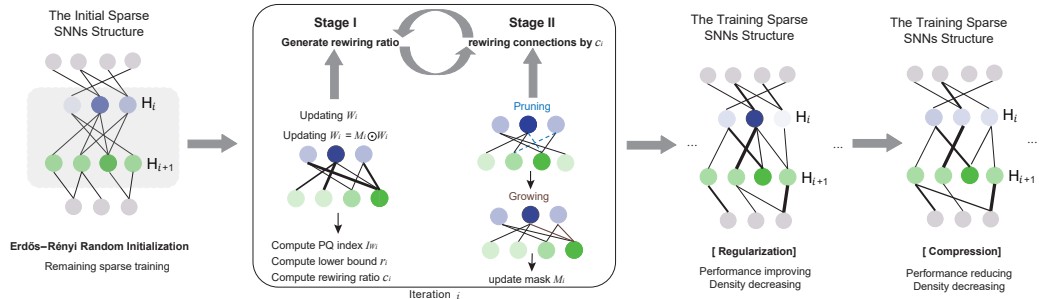

Figure 1: The flowchart of the proposed two-stage sparse structure learning method for SNNs. Stage I involves the typical training process and attempts to identify an appropriate rewiring ratio according to PQ index. Stage II conducts the dynamic sparse structure learning method based on the rewiring ratio in stage I. The iterative learning of the above two stages is employed during the whole training process, thereby implementing the sparse training from scratch for SNNs and enhancing generalization ability of the sparse model.

tionality while reducing computational redundancy, thus achieving superior compression of SNN architectures.

## 2 RELATED WORKS

SNNs have seen considerable advancements in learning algorithms that have expanded their parameter capacity and diversified their topological structures. These networks often incorporate established ANNs architectures, including VGG11, ResNets19, and Transformers, adapting them to the spike-based processing paradigm (Yao et al., 2024; Hu et al., 2024). Training methodologies range from direct training with surrogate gradients to conversion techniques that transform pre-trained ANNs into SNNs. While these static-topology SNNs have demonstrated significant efficacy in various applications, such as object detection and natural language understanding, they primarily emphasize synaptic weight optimization (Gast et al., 2024; Zheng et al., 2024; Ren et al., 2024). This focus tends to overlook the critical aspect of synaptic connectivity learning, frequently leading to parameter inefficiencies and constrained network evolution. In contrast, SNNs designed with dynamic structures learning are engineered to concurrently optimize both synaptic connections and weights. This dual optimization affords enhanced flexibility and facilitates the development of more efficient and adaptive network topologies (Xu et al., 2024; Jiang et al., 2023; Shen et al., 2024). We categorize the current sparse structure learning methods for SNNs into two distinct groups.

**Gradual Sparsification of Connection Structures for SNNs.** This kind of methods typically initializes the network with a non-sparse connected structure, which is iteratively optimized throughout training, resulting in a gradually sparser connection structure. 1) Weight parameter optimization methods. For instance, the gradient rewiring (Grad R) method is introduced in (Chen et al., 2021) which implements sparse structure learning through redefining network connection parameters. This method ensures that the gradient of these parameters forms an angle of less than 90° with the accurate gradient. During model training, synaptic pruning and regeneration are iteratively applied, achieving joint learning of synaptic connections and weights. Building on this, the nonlinear gradient reparameterization function that controls pruning speed through a threshold growth function is introduced in (Chen et al., 2022), further optimizing the SNNs structure. (Shi et al., 2023) combines unstructured weight pruning with unstructured neuron pruning to maximize the utilization of the sparsity of neuromorphic computing, thereby enhancing energy efficiency. 2) Regularization-based methods. (Deng et al., 2021a) incorporated gradient regularization into the loss function, achieving synaptic connection pruning and weight quantization based on the Alternating Direction Method of Multipliers (ADMM). Similarly, Yin et al. combined sparse spike encoding with sparse network connections, using sparse regularization to establish models for spike data transmission and network sparsification (Yin et al., 2021). There are also some studies to explore the connection pruning for spiking-based Transformer structure (Liu et al., 2024a). 3) Connection-relationship-determination-

based methods. The synaptic sampling method based on Bayesian learning is proposed in (Kappel et al., 2015), modeling dendritic spine movement characteristics to achieve synaptic connection reconstruction and weight optimization. Combining unsupervised STDP rules with supervised Tempotron training, SNNs with connection gates are developed in (Qi et al., 2018). It improves the performance while reducing connections via sparse SNNs. There are also other studies to introduce the plasticity-based pruning methods for deep SNNs (Han et al., 2024b).

**Fully Sparsification of Connection Structures for SNNs.** A different strategy involves initializing the network with a sparse connection structure from the start and continually optimizing this sparse structure throughout training. This fully sparse training approach is particularly advantageous for hardware implementation in resource-constrained environments, such as on-chip training in hardware chips. 1) Synaptic connection-rewiring-based methods. These evolutionary structure learning methods are proposed for deep SNNs by drawing inspiration of rewiring mechainism in human brain (Han et al., 2024a; Shen et al., 2023; Li et al., 2024b). This method employs synaptic growth and pruning rules to adaptively adjust the connection structure based on gradients, momentum, or amplitude during training, maintaining a certain level of sparsity in synaptic connections and achieving effective sparse training of SNNs. 2) Lottery-ticket-hypothesis-based methods. The architecture search could also generate sparse SNNs, such as the lottery ticket hypothesis. The Early-Time lottery ticket hypothesis method proposed in (Kim et al., 2022) demonstrates that winning sparse subnetworks exist in deep SNNs, similar to traditional deep ANNs. Further, the utilization-aware LTH method, which incorporates intra-layer connection regeneration and pruning during training, addresses hardware load imbalance issues caused by unstructured pruning methods (Yin et al., 2024).

Despite these advancements, a gap remains in the deployment of fully adaptive and efficient SNNs architectures, particularly in resource-constrained environments such as edge computing devices. This underscores the necessity for novel methods that not only refine the sparsity and efficiency of these networks but also maintain adaptive learning capabilities throughout their lifecycle during the whole training process. The need for dynamic, flexible SNN models that mirror the human brain's ability to reorganize and optimize its neural pathways in real-time is clear.

Our research addresses this gap by proposing a two-stage dynamic sparse structure learning approach for SNNs from scratch, leveraging the latest advances in neural network compression and synaptic plasticity. This method promises to significantly enhance the adaptability and efficiency of deep SNNs, positioning them as a viable solution for next-generation neuromorphic computing Ma et al. (2024), particularly as future highly efficient perceptron applications (Chen et al., 2025; Guan & Zhao, 2022; Guan et al., 2024; Liang et al., 2024) increasingly demand more computational resources for higher performance. We believe that by integrating adaptive synaptic pruning and growth mechanisms, our approach will set a new standard for sparse structure learning in SNNs, aligning closely with the natural efficiencies observed in biological neural processes.

## 3 METHODS

The main goal of our study is to implement the fully sparse training from scratch for SNNs with the dynamic compressibility during the training process. In detail, we first introduce the proposed two-stage sparse learning framework for SNNs. After that, the first and second stage computations for obtaining the right rewiring ratio and rewiring sparse networks are described, respectively.

### 3.1 THE FRAMEWORK OF THE TWO-STAGE SPARSE LEARNING METHOD

As illustrated in Fig. 1 and Algorithm 1, we design the two-stage sparse training method for SNNs with an appropriate rewiring ratio for each iteration during the training process. The sparse weight connections are initialized according to the Erdös–Rényi (ER) Random Graph. The ER graph could guarantee that the synaptic connection for each neuron has the same connection probability. Assuming there are $n^k$ and $n^{k-1}$ neurons in the neighboring two layers, then the probability of weight connection mask $M_{k,k-1} = 1$ between two neurons in these two layers satisfies

$$p(M_{k,k-1} = 1) = \frac{\epsilon(n^k + n^{k-1})}{n^k * n^{k-1}}. \tag{1}$$

where $\epsilon$ is a constant (or scaling factor) that influences the edge probability and accounts for sparsity or connectivity scaling. Then the corresponding weight value $W$ can be initialized by the commonly

---

**Algorithm 1** The two-stage sparse training process of SNNs.

---

**Input Data**: $x_i, i = 1, 2, ..., N$.
**Labels of Input Data**:$y_i, i = 1, 2, ..., N$.
**Parameters**: The weight mask is $M$. The weight matrix: $W$. The updating iterations: $Epoch_{frequency}$.

1: **for** each assigned sparse layer of the SNNs **do**
2:     Initialize the sparse weight mask of the connected layer as the Erdös–Rényi topology;
3: **end for**
4: Initialize trained weight parameters;
5: **for** each training iterations $i$ **do**
6:     $\diamond$ **Stage I**
7:     Perform standard training procedure with $W_i = M_i \odot W_i$;
8:     Perform weights updates for $W_i$;
9:     Compute the total number of model parameters $d_i = \mid M_i \mid$;
10:     Compute PQ Index $I_{W_i}$ and the lower bound of the amount of remaining parameters $r_i$;
11:     Compute the rewiring ratio $c_i$;
12:     $\diamond$ **Stage II**
13:     **if** current training epoch $\% Epoch_{frequency} == 0$: **then**
14:         **for** each assigned sparse layer of SNNs **do**
15:             Remove the fraction $c_i$ of synaptic connections according to the pruning rule;
16:             Regrow the fraction $c_i$ of synaptic connections according to the growing rule;
17:         **end for**
18:     **end if**
19: **end for**
20: **return** The sparse SNNs with $W$.

---

used initialization method, such as Xavier Initialization and random normal distribution Initialization. Since then, we have been able to obtain the initialized sparse SNNs. After that, the initialized SNNs would be trained over multiple iterations, through the iterative training of the first stage and second stage in each iteration. It is worth noting that the SNNs would remain sparse and dynamically search for the suitable rewiring ratio in the following training process.

In detail, the first stage involves the typical training process and attempts to identify an appropriate rewiring ratio based on temporarily trained weights. The rewiring ratio is calculated according to the PQ index, an efficient measure of the compressibility of neural network models (Diao et al., 2023). The PQ index helps quantify the redundancy in the network, thereby informing the following rewiring strategy. In the second stage, the dynamic sparse structure learning method based on the rewiring method is adopted to implement sparse training from scratch. The connections are iteratively pruned and regrown according to the specified rewiring ratio. This iterative training approach ensures that the network continuously adapts and optimizes its structure, thereby improving performance. The rewiring method allows for dynamic adjustment, promoting the activation and growth of previously dormant connections, which contributes to SNNs' capability enhancement.

By integrating these two stages, our method achieves efficient and effective sparse training for SNNs, leveraging the compressibility insights gained in the first stage to guide dynamic structural adjustments in the second stage. This approach not only maintains the sparsity and efficiency of the model but also enhances its generalization ability.

### 3.2 COMPRESS THE SPARSE SNNS BASED ON PQ INDEX

After the sparse initialization based on ER graph, the synaptic connections between neurons would become sparse randomly. Then in the following training process, the SNNs model would be trained according to the two stages sparse training method.

In the first stage, we train the sparse SNNs and compute the appropriate rewiring ratio according to PQ index $I_{p,q}(W)$ (we simplified it as $I(W)$ in the detailed derivation in the supplementary materials). Here is the derivation of the sparsity measure $I_{p,q}(W) = 1 - d^{\frac{1}{q} - \frac{1}{p}} \cdot \frac{\|W\|_p}{\|W\|_q}$ for spiking neural

networks (SNNs), incorporating the formula update and focusing on scaling invariance, sensitivity to sparsity reduction, and cloning invariance, combined with spatiotemporal dynamics and sparsity in SNNs. In detail, the scaling invariance in SNNs corresponds to: (1) Independence of weight scaling: If the weight matrix $W$ is scaled (e.g., multiplied by a constant), its sparsity structure remains unchanged, and so should $I_{p,q}(W)$. (2) Independence of temporal scaling: Changes in spike magnitudes (the activation value) should not affect the sparsity measure, ensuring the measure accurately reflects temporal dynamics. We give detailed derivation in SNNs, it ensures that $I_{p,q}(W)$ remains unaffected when all weights are scaled proportionally (e.g., multiplying $W$ by a constant $\alpha > 0$). The scaling weight magnitudes or activation value intensity does not change the network sparsity. Meanwhile, we analyze that $I_{p,q}(W)$ keeps sensitivity to spatial and temporal sparsity in SNNs, that is, the distribution of weights or spike activations (firing rates). When it changes weight distribution with more nonzero weights, leading to a reduction in $I_{p,q}(W)$ corresponds to sparsity decreasing. When temporal sparsity decreases (more neurons firing at the same time), the distribution becomes denser, which directly affects the ratio $\|W\|_p/\|W\|_q$, leading to a decrease in $I_{p,q}(W)$. In addition, we demonstrate that the sparsity measure $I_{p,q}(W)$ should remain invariant when the weight matrix is cloned in spatial and temporal dimensions. This ensures that cloning or repeating the matrix does not affect the sparsity measure.

In detail, the PQ index for the non-zero vector $W_i \in \mathbb{R}_d$ with any $0 < p < q$ is computed by:

$$I_{p,q}(W_i) = 1 - d^{\frac{1}{q}-\frac{1}{p}}(\| W_i \|_p - \| W_i \|_q), \tag{2}$$

where $\| W_i \|_p$ equals to $(\sum_{j=1}^{d} | w_j |^p)^{1/p}$, in which $w_j, j = 1...d$ is the non-zero element in $W_i$. Then the lower bound of the retaining number of model parameter $W_i$ can be obtained by:

$$r_i = d_i(1 + \alpha_r)^{-q/(q-p)}[1 - I_{p,q}(W_i)]^{pq/(q-p)}, \tag{3}$$

where $d_i = |M_i|$. Then the pruned ratio with better compressibility is computed by:

$$c_i = \lfloor d_i \cdot min(\gamma(1 - \frac{r_i}{d_i}), \beta)\rfloor/N_{W_i}, \tag{4}$$

in which the $\gamma$ and $\beta$ are the scaling factor and maximum rewiring ratio, respectively. In detail, the hyperparameter of $\gamma$ is used to scale the rewiring ratio according to PQ index. The bigger $\gamma$ would obtain the higher rewiring ratio. We follow the settings in (Diao et al., 2023) to set $\gamma = 1$ and $\beta = 0.9$ to prevent the model are over-pruned seriously. $N_{W_i}$ is the total number of parameters in $W_i$. Assume $r$ is the indices set of $W_i$ with the largest weight magnitude. Then $\alpha_r$ denotes the smallest value satisfying $\sum_{j \notin M_i^r} |w_j|^p \leq \alpha_r \sum_{j \in M_i^r} |w_j|^p$. The big $\alpha_r$ implies the model parameters are redundant and would result in a higher rewiring ratio. Thus we set the $\alpha_r$ to be 0.001 in the experiments to slow down the pruning speed and improve the stability of sparse model training. We give an example in Fig. 2 (a).

Since then, the right rewiring ratio for weight parameters in sparse SNNs has been figured out to improve compressibility and prevent sparse SNNs from either over-pruning or under-pruning (Li et al., 2024a; Xu et al., 2024).

During the training process in each iteration, before the above spasity measurement process, we need to update the weight matrix according to the learning algorithm in our model. In this paper, we adopt the iterative Leaky Integrate-and-Fire (LIF) neuron model in SNNs to enhance information integration and temporal representation (Wu et al., 2019). The membrane potential $u(t)$ of postsynaptic neuron is updated based on the membrane potential at $t-1$ and the integrated presynaptic neuron input:

$$u(t) = \tau u(t-1) + (M_i \odot W_i)x(t), \tag{5}$$

where $\tau$ is the leaky factor set to 0.5, and $x(t)$ represents the spike inputs. When $u(t)$ exceeds the firing threshold of $V_{th}$, the neuron fires a spike, and $u(t)$ is set to be 0. Consequently, the neuron output and the membrane updating are given by:

$$a(t+1) = \Theta(u(t+1) - V_{th}), \tag{6}$$

$$u(t+1) = u(t+1)(1 - a(t+1)), \tag{7}$$

To ensure that the output signal at each time step approximates the target distribution, we utilize the temporal efficient training loss function as in (Deng et al., 2021b):

$$L_{TET} = \frac{1}{T}\sum_{t=1}^{T} L_{CE}[O(t), y], \tag{8}$$

where $T$ denotes the time steps and $L_{CE}$ is the cross-entropy loss function. Then the masked weights are updated according to the gradient descent rule with surrogate gradient function.

### 3.3 THE CONNECTION REWIRING OF DYNAMIC SPARSE STRUCTURE LEARNING

After the above stage obtains the appropriate rewiring ratio, the connection rewiring method is followed to improve the stability and generalization ability of the sparse SNNs, instead of pruning the weights with the smallest magnitudes directly. The connection rewiring method is motivated by the synaptic rewiring mechanism in the human brain. The synaptic rewiring in the brain, covering the processes of synaptic pruning (elimination) and synaptic growth (formation), plays a vital role in neural development, learning, memory, and overall cognitive function. This dynamic remodeling of synaptic connections promotes the brain's adaptability and efficiency in processing information. Meanwhile, the effectiveness of the rewiring operation has been demonstrated in earlier works for the structure learning of SNNs. Therefore, we employ the dynamic connection rewiring process to implement the sparse training of SNNs models from scratch, thus improving the stability and generalization ability of the sparse SNNs.

The connection rewiring method could implement the effective and fast training by the iterative training of pruning and regrowing connections, avoiding the introduction of additional parameters that could increase memory usage. The pruning rule ensures the elimination of less significant connections in SNNs, reducing computational complexity while preserving the core structure of the network. We rank the weights magnitude according to their absolute values in sparse SNNs trained at the first stage, and prune the weight connections with the rewiring ratio of $c_i$ in the above section. However, the only operation of pruning may destroy the stable convergence of sparse SNNs and restrict the network's expressive capacity. To fully leverage the information processing capacity of the original large SNNs without any pruning, it is crucial that all connections are activated during training. Consequently, the growth rule is employed to promote the regeneration of connections that have not been activated for a significant period of time. Different pruning and regrow rules adapt to the proposed two-stage sparse structure learning framework. For simplify, we adopt the momentum-based growing rule, to prioritize the regeneration of synaptic connections according to the momentum of the parameters. This approach ensures that connections showing significant momentum, and thus potential importance, are prioritized for regrowth.

In detail, after obtained the suitable rewiring ratio in the first stage, we could then use this ratio $c_i$ to guide the training in the second stage. Assuming the standard training procedure for sparse SNNs is followed by freezing the masked weights as $W_i = M_i \odot W_i$, where the mask $M_i$ is obtained according to the pruning and regrowing principle as mentioned above. After obtaining the updated $W_i$, we begin to compute the rewiring ratio again as described in the first stage. Therefore, through the iterative process between the first and second stage, our model can improve the efficiency of sparse training SNNs from scratch with the adaptive rewiring ratio.

## 4 EXPERIMENTS

In this section, we evaluate the performance of our proposed two-stage sparse structure learning method for SNNs. We conduct experiments on the CIFAR10, CIFAR100, and DVS-CIFAR10 datasets, including both ablation studies and comparative experiments. The experiments environments are NVIDIA-4090 GPU computation devices based on PYTORCH framework.

### 4.1 ANALYSIS ON THE DYNAMIC SPARSE TRAINING DURING TRAINING PROCESS

The performances of the proposed two-stage dynamic sparse training of SNNs are validated on two different rewiring scopes, including neuron-wise rewiring and layer-wise rewiring. The neuron-wise rewiring would adopt the connection rewiring each neuron of model parameters, which would prune and regrow $d \cdot p$ connected weight parameters for each neuron, and the rewiring ratio of each neuron is computed by the PQ index respectively. While the layer-wise one conducts weight parameter rewiring for each layer separately.

**Effectiveness in the layer-wise rewiring scope.** The accuracy and connection density of the proposed two-stage sparse training method for SNNs are illustrated in Fig. 3. The initial connection

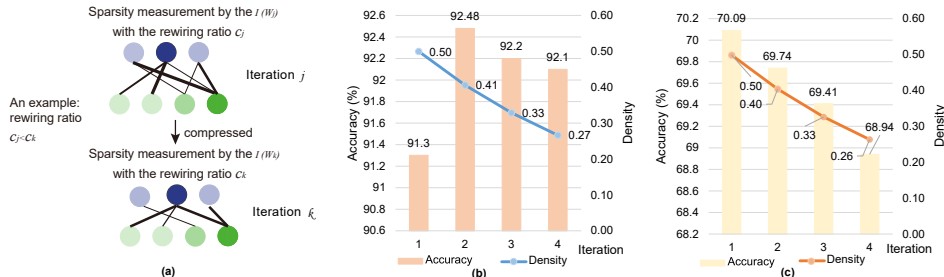

Figure 2: An example for training process of our method (a). The performance of the proposed two-stage sparse training method for SNNs, on CIFAR10 (b) and CIFAR100 (c) datasets, in the neuron-wise pruning scope. The bar chart represents the accuracy achieved by the proposed two-stage sparse training method. The solid line reflects the density of synaptic connections in the SNNs model.

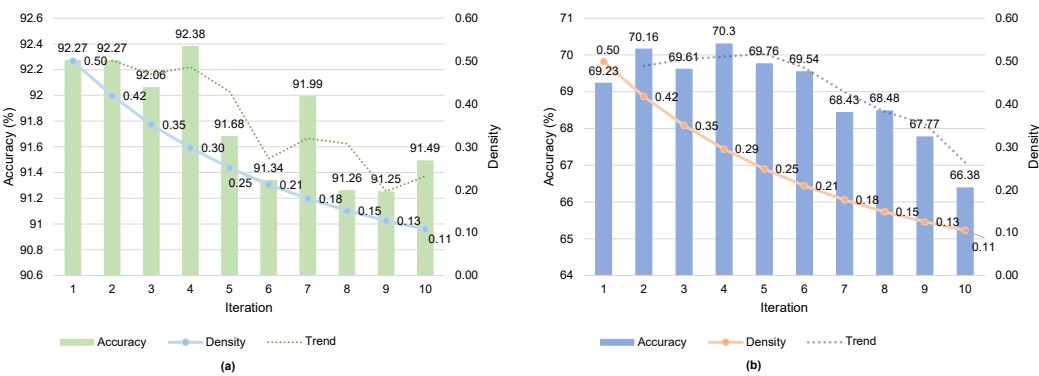

Figure 3: The performance of the proposed two-stage sparse training method for SNNs, on CIFAR10 (a) and CIFAR100 (b) datasets, in the layer-wise pruning scope. The bar chart represents the accuracy achieved by the proposed method. The solid line reflects the density of synaptic connections in the SNNs model. The dashed line is the trend analysis of accuracy using a two-period moving average. This diagram depicts the correlation between the density of the model and the enhancements in performance achieved using our two-stage sparse structure learning technique.

density is set to be 0.5, which means that there are only half of the connections to be activated when initialization. Meanwhile, the connections in our model remain sparse, ranging from 0.5 to 0.11 during the whole training process. In addition, as the number of iterations in the sparse training process increases, the proposed two-stage sparse training method generates a relatively suitable rewiring ratio in the first stage, gradually reducing the synaptic connection density in SNNs. It is notable that the proposed model achieves its peak accuracy of 92.38% and 70.3% during the fourth iteration with a connection density of 30% on the CIFAR10 and CIFAR100 datasets, respectively. The peak accuracy is even higher than that (about 92.2% on the CIFAR10 dataset) of densely connected SNNs. This improvement can be attributed to the connection rewiring, which introduces a more activated parameter space and enhances the performance of sparse training by exploring extensive parameters throughout the sparse training process.

Simultaneously, the overall accuracy of the sparse SNNs exhibits a fluctuating trend. In the initial iterations, our model's stage I produces appropriate levels of sparsity, which decreases the density of synaptic connections in the sparse SNNs while improving accuracy. However, as the iterations continue and the model becomes more compressed, the performance starts to decline moderately due to increased sparsity. At a crucial point, when the model achieves its highest level of accuracy during the fourth iteration with a connection density of 30% for CIFAR10, additional pruning results in a collapse when important parameters are eliminated, leading to considerable performance decrease.

**Effectiveness in the neuron-wise rewiring scope.** We also verify the performance of our proposed two-stage sparse straining method in the neuron-wise scope. As illustrated in Fig. 2, we analyze the accuracy and the corresponding density within four iterations, for these four iterations have shown the main trend change as in the situation of layer-wise scope. As shown in Fig. 2, the proposed model exhibits similar accuracy oscillation phenomena in the CIFAR10 dataset when using layer-wise sparse structure training, akin to the neuron-wise approach. The proposed model achieves an accuracy of 92.48% at the second iteration with a connection density of only 41%. However, in the neuron-based scenario on the CIFAR100 dataset, no similar oscillation phenomena are observed. This could be due to the initial high sparsity, which may have led to the pruning of some critical synaptic connections. Thus, the remaining connections could not be insufficiently trained, resulting in decreased performance as the connection densities reduce.

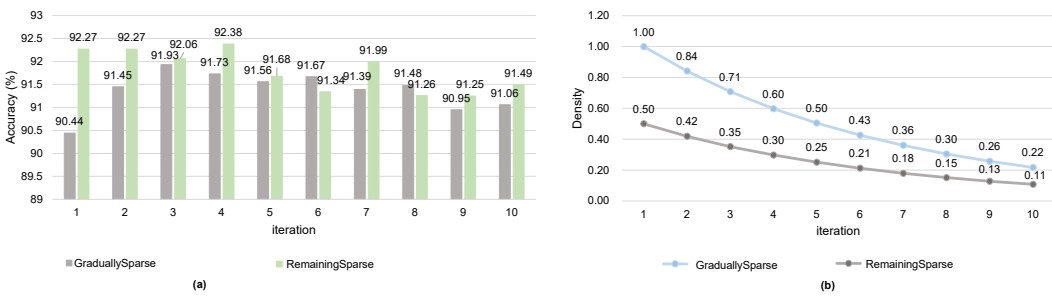

Figure 4: The ablation experiments of our proposed two-stage sparse training method for SNNs. (a) The accuracy comparison between the gradually sparse training and sparse training from scratch. (b) The connection density comparison between the gradually sparse training and sparse training from scratch.

The above phenomena are consistent with the proposed model's behavior. The pruning process initially removes redundant parameters. This results in a decrease in the sparsity of model parameters, leading to an improvement in performance due to regularization effects. As the model is compressed more, some critical parameters are pruned when the model reaches convergence, resulting in an increase in sparsity and a slight decrease in performance. Eventually, as the model begins to collapse, all weakened parameters are removed, leaving only the essential parameters needed to sustain performance. Consequently, the level of sparsity decreases dramatically, leading to a noticeable decline in performance.

**Ablation study on the sparse training from scratch.** To evaluate the effectiveness of sparse training for the proposed two-stage sparse structure learning method, we conduct the ablation study by comparing the performance with gradually sparse training and sparse training from scratch. As illustrated in Fig. 4, the performance of the sparse training from scratch (Remaining Sparse) outperforms that of the gradually sparse training from the initial fully non-sparse connections. The reason lies in that the manner of the sparse training from scratch explores a similar thorough parameter space to the non-sparse model and masks some noises caused by the redundancy parameters. Besides, the proposed model for sparse training from scratch demonstrates superior network connection sparsity than the gradually sparse training model at the same number of iterations. This allows the sparse training from scratch model to more quickly identify the optimal rewiring rate and achieve better performance. Additionally, the sparse training from scratch model is more hardware-friendly than the gradually sparse training model, making it more suitable for sparse training in hardware-constrained environments especially on-chip learning.

## 4.2 PERFORMANCE COMPARISON TO OTHER METHODS

We compare the performance of the proposed two-stage sparse structure learning method for SNNs with other current state-of-the-art SNNs: ADMM (Deng et al., 2021a), Grad R (Chen et al., 2021), ESLSNN (Shen et al., 2023) and STDS (Chen et al., 2022), UPR (Shi et al., 2023).

As shown in Tab. 1, the proposed two-stage sparse structure training method achieves competitive performance among the various methods while retaining the advantage of sparse training from

Table 1: Performance comparison of the proposed two-stage sparse structure learning approach for SNNs with other models.

| Dataset | Pruning Method | Architecture | T | Top-1 Acc.(%) | Acc. Loss(%) | Conn. (%) | Param. (M) | SOPS (M) |
|---|---|---|---|---|---|---|---|---|
| CIFAR10 | ADMM | 7 Conv, 2 FC | 8 | 90.19 | -0.13 | 25.03 | 15.54 | - |
| | Grad R | 6 Conv, 2 FC | 8 | 92.54 | -0.30 | 36.72 | 10.43 | - |
| | ESLSNN | ResNet19 | 2 | 91.09 | -1.7 | 50 | 6.3 | 180.56 |
| | STDS | 6 Conv, 2 FC | 8 | 92.49 | -0.35 | 11.33 | 1.71 | 147.22 |
| | UPR | 6 Conv, 2 FC | 8 | 92.05 | -0.79 | 1.16 | 9.56 | 16.47 |
| | **This work** | **ResNet19 Neuron-wise** | **2** | **92.48** | **+1.18** | **40.58** | **5.12** | **158.35** |
| | | | | **92.1** | **+0.8** | **26.63** | **3.36** | **121.49** |
| | | **ResNet19 Layer-wise** | **2** | **92.38** | **+0.11** | **29.72** | **3.7** | **133.26** |
| | | | | **91.99** | **-0.28** | **17.91** | **2.26** | **110.65** |
| CIFAR100 | ESLSNN | ResNet19 | 2 | 73.48 | -0.99 | 50 | 6.32 | 186.25 |
| | UPR | SEW ResNet18 | 4 | 70.45 | -3.71 | 3.60 | - | 9.60 |
| | | | | 69.41 | -4.75 | 2.48 | | 6.79 |
| | **This work** | **ResNet19 Layer-wise** | **2** | **70.3** | **+1.07** | **29.48** | **3.73** | **140.27** |
| DVS-CIFAR10 | ESLSNN | VGGSNN | 10 | 78.3 | -0.28 | 10 | 0.92 | 129.64 |
| | STDS | VGGSNN | 10 | 79.8 | -2.6 | 4.67 | 0.24 | 38.85 |
| | UPR | VGGSNN | 10 | 78.3 | -0.5 | 0.77 | 1.81 | 6.75 |
| | | | | 81.0 | -1.4 | 4.46 | 2.5 | 31.86 |
| | **This work** | **VGGSNN Layer-wise** | **10** | **78.4** | **+0.08** | **30** | **2.76** | **189.02** |

scratch. Notably, compared to fully non-sparse models, our sparse training model can even improve performance while maintaining a certain level of sparsity through dynamic iteration and searching for an appropriate rewiring ratio. For example, on the CIFAR10 dataset, the model trained using our proposed method under the neuron-wise scope improves performance by approximately 1% compared to the fully non-sparse model while maintaining a sparsity of 30% to 40%. On the CIFAR100 dataset, the performance of SNNs model with our two-stage sparse training method is also improved 1.07% compared to the non-sparse model with only 29.48% connection. It is worth noting that the proposed two-stage training method proceeds sparse training from scratch and maintains sparse training during the whole training process. These results demonstrate that our proposed model helps the original non-sparse model mask redundant parameters and enhance the generalization capability of the sparse model during iterative training by continuously finding the appropriate rewiring ratio.

## 5 CONCLUSION

To summarize, this study has introduced a novel two-stage dynamic structure learning method tailored for SNNs that effectively addresses the challenges of fixed pruning ratios and the limitations of static sparse training methods prevalent in current models. In the first stage of our strategy, we employ the PQ index to evaluate the compressibility of sparse subnetworks. This enables us to make informed adjustments to the rewiring ratios of synaptic connections. This adaptive technique enables the model to circumvent the drawbacks of insufficient pruning or excessive pruning. In the second stage, the predetermined rewiring ratios guide the dynamic synaptic connection rewiring, incorporating both pruning and regrowth strategies. This approach not only improves the compression efficiency of sparse SNNs but also boosts their performance. The iterative learning process implemented across both stages ensures continuous improvement and adaptation of the sparse network structure throughout the training phase. The experimental results validate that the proposed dynamic structure learning greatly improves the compression efficiency of SNNs. Additionally, it either matches or exceeds the performance benchmarks set by current models in certain circumstances. Crucially, this strategy maintains the benefits of sparse training from the scratch, which is particularly advantageous in settings with restricted hardware resources, like neuromorphic hardware on Edge AI.

## 6 ACKNOWLEDGEMENT

This work was supported by National Natural Science Foundation of China under Grant (No. 62306274, 62088102, 62476035, 62206037, 61925603), Open Research Program of the National Key Laboratory of Brain-Machine Intelligence, Zhejiang University (No. BMI2400012).

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
