# A    Sparsity in Spiking Neural Networks (SNNs) and the Derivation of $I(W)$

Here is the derivation of the sparsity measure $I(W) = 1 - d^{1/q-1/p} \cdot \frac{\|W\|_p}{\|W\|_q}$ (which denotes as $I_{p,q}(W)$ in the manuscript) for spiking neural networks (SNNs), incorporating the formula update and focusing on scaling invariance, sensitivity to sparsity reduction, and cloning invariance, combined with spatiotemporal dynamics and sparsity in SNNs.

SNNs communicate through discrete spikes, exhibiting the following key features: (1) Discrete activation: Postsynaptic neurons emit spikes only at specific time points. They are either active (firing spikes) or inactive (not spiking), resulting in sparse data flow. (2) Structure sparsity: Sparsity refers to the proportion of nonzero elements in a weight matrix.

The sparsity measure $I(W) = 1 - d^{1/q-1/p} \cdot \frac{\|W\|_p}{\|W\|_q}$ is rigorously constructed to reflect these properties. The sparsity measure $I(W) = 1 - d^{1/q-1/p} \cdot \frac{\|W\|_p}{\|W\|_q}$ satisfies these properties, where: $\|W\|_p = \left(\sum_{i=1}^{d} |w_i|^p\right)^{1/p}$ is the $\ell_p$-norm of $W$, $\|W\|_q = \left(\sum_{i=1}^{d} |w_i|^q\right)^{1/q}$ is the $\ell_q$-norm of $W$, $d$ is the dimensionality of $W$, $p < q$ ensures that sparsity is more effectively captured. The additional term $1-$ allows $I(W)$ to range between 0 (no sparsity) and 1 (maximum sparsity). Because when the sparsity is 100%, which means all the elements in SNNs are 0, then $I(W)$ is 1. While when there are no zero elements, that is, the orginal fully connected SNNs, then $I(W)$ is 0. The term $d^{1/q-1/p}$ ensures that $I(W)$ is independent of the vector length, satisfying the cloning property. Without this term, it would vary with the size of $W$, even for identical sparsity patterns. Below, we derive this formula and explain how it aligns with SNN characteristics.

In detail, the measure $I(W)$ is designed to satisfy the following key properties:

## A.1    Scaling Invariance

In SNNs, the scaling invariance corresponds to: (1) Independence of weight scaling: If the weight matrix $W$ is scaled (e.g., multiplied by a constant), its sparsity structure remains unchanged, and so should $I(W)$. (2) Independence of temporal scaling: Changes in spike magnitudes (the activation value) should not affect the sparsity measure, ensuring the measure accurately reflects temporal dynamics.

Under the constraints of sparsity measurement, the sparsity measure should remain unchanged if the weight matrix $W$ is scaled by a positive constant $\alpha > 0$. Specifically:

$$I(\alpha W) = 1 - d^{1/q-1/p} \cdot \frac{\|\alpha W\|_p}{\|\alpha W\|_q},$$

Since:

$$\|\alpha W\|_p = \alpha \|W\|_p \quad \text{and} \quad \|\alpha W\|_q = \alpha \|W\|_q,$$

substituting into $I(W)$ yields:

$$I(\alpha W) = 1 - d^{1/q-1/p} \cdot \frac{\alpha \|W\|_p}{\alpha \|W\|_q} = 1 - d^{1/q-1/p} \cdot \frac{\|W\|_p}{\|W\|_q} = I(W).$$

Therefore, in SNNs, it ensures that $I(W)$ remains unaffected when all weights are scaled proportionally (e.g., multiplying $W$ by a constant $\alpha > 0$). Meanwhile, a natural advantage lies in the fact that SNNs rely solely on discrete spike timing and firing rates to transmit information, ensuring consistency across all magnitudes of discrete spike trains. Therefore, the scaling weight magnitudes or activation value intensity do not change the network sparsity.

## A.2    Sensitivity to Sparsity Reduction

In SNNs, sparsity reduction can occur in two distinct forms: (1) Weight sparsity: Decreased sparsity corresponds to more nonzero weights, leading to a reduction in $I(W)$. (2) Temporal sparsity: If more neurons fire simultaneously, temporal sparsity decreases, and $I(W)$ reflects this reduction.

Consider two weight matrices: (1) $W_1 = [10, 0, 0, 0]$. The sparse one with few neurons fire, resulting in a smaller $\|W\|_p$, a lower $\|W\|_q$, and a high $I(W)$. (2) $W_2 = [5, 5, 0, 0]$. Less sparse one with more neurons fire simultaneously, increasing $\|W\|_p$ more then $\|W\|_q$, causing $I(W)$ to decrease compared to the case with $W_1$.

1. Compute norms: - $\|W_1\|_p = 10$, $\quad \|W_2\|_p = 2^{1/p} \cdot 5$, $\quad \|W_1\|_q = 10$, $\quad \|W_2\|_q = 2^{1/q} \cdot 5$

2. Sparsity measure:

$$I(W_1) = 1 - d^{1/q - 1/p}, \quad I(W_2) = 1 - d^{1/q - 1/p} \cdot 2^{1/p - 1/q}.$$

3. Since $p < q$, $1/p - 1/q > 0$, so $2^{1/p - 1/q} < 1$. Thus:

$$I(W_2) < I(W_1).$$

Thus, it keeps sensitivity to spatial and temporal sparsity, that is, the distribution of weights or spike activations (firing rates). When it changes weight distribution with more nonzero weights, leading to a reduction in $I(W)$ corresponds to sparsity decreasing. When temporal sparsity decreases (more neurons firing at the same time), the distribution becomes denser, which directly affects the ratio $\|W\|_p/\|W\|_q$, leading to a decrease in $I(W)$.

### A.3 CLONING INVARIANCE

It should satisfy the property of Cloning Invariance in SNNs from these two aspects: (1) Spatial network expansion: Cloning weights for larger networks does not change sparsity. (2) Temporal expansion: Repeating activities over time does not affect sparsity, ensuring temporal consistency.

For the case of incorporating spatial vectors, the sparsity measure $I(W)$ should remain invariant when the weight matrix is cloned:

$$I(W) = I([W, W])$$

This ensures that cloning or repeating the matrix does not affect the sparsity measure.

1. For a cloned matrix $[W, W]$:

$$\|[W, W]\|_p = 2^{1/p}\|W\|_p, \quad \|[W, W]\|_q = 2^{1/q}\|W\|_q$$

2. Substituting into $I([W, W])$:

$$I([W, W]) = 1 - (2d)^{1/q - 1/p} \cdot \frac{\|[W, W]\|_p}{\|[W, W]\|_q}$$

$$I([W, W]) = 1 - (2d)^{1/q - 1/p} \cdot \frac{2^{1/p}\|W\|_p}{2^{1/q}\|W\|_q}$$

3. Simplify:

$$I([W, W]) = 1 - d^{1/q - 1/p} \cdot \frac{\|W\|_p}{\|W\|_q} = I(W)$$

For the case of incorporating time steps in SNNs, if $W$ is repeated across $T$ time steps:

$$W_T = [W, W, \ldots, W] \in \mathbb{R}^{d \times (nT)}.$$

1. Norms for $W_T$:

$$\|W_T\|_p = T^{1/p}\|W\|_p, \quad \|W_T\|_q = T^{1/q}\|W\|_q$$

2. Sparsity measure:

$$I(W_T) = 1 - (nT)^{1/q - 1/p} \cdot \frac{\|W_T\|_p}{\|W_T\|_q}$$

3. Substituting:

$$I(W_T) = 1 - (nT)^{1/q - 1/p} \cdot \frac{T^{1/p}\|W\|_p}{T^{1/q}\|W\|_q}$$

4. Simplify:

$$I(W_T) = 1 - n^{1/q - 1/p} \cdot \frac{\|W\|_p}{\|W\|_q} = I(W)$$

In addition, considering temporal sparsity changes, if neuron activity differs across time steps, sparsity decreases. For dynamic weights $W_T' = [W^{(1)}, W^{(2)}, \ldots, W^{(T)}]$, norms reflect this change:

$$\|W_T'\|_p = \left( \sum_{t=1}^{T} \|W^{(t)}\|_p^p \right)^{1/p}, \quad \|W_T'\|_q = \left( \sum_{t=1}^{T} \|W^{(t)}\|_q^q \right)^{1/q}$$

Sparsity measure decreases with reduced temporal sparsity:

$$I(W_T') = 1 - (nT)^{1/q - 1/p} \cdot \frac{\|W_T'\|_p}{\|W_T'\|_q}$$

Therefore, it satisfies the property of cloning invariance in SNNs from the spatial and temporal dimensions.

## A.4 Sparsity Decreases as More Neurons Fire

When the values of weights in SNNs are adjusted such that more neurons are active (e.g., more neurons spike simultaneously), the sparsity should decrease.

The proof is here, in SNNs, the activation pattern of neurons is sparse. When more neurons fire simultaneously, the weight matrix becomes denser (i.e., fewer zero entries in the matrix). This means that temporal sparsity is reduced, and more activations lead to a lower value for the sparsity measure $I(W)$. As the number of neurons firing simultaneously increases, $\|W\|_p$ grows faster than $\|W\|_q$, which causes $\frac{\|W\|_p}{\|W\|_q}$ to increase, and thus $I(W)$ decreases.

## A.5 Neural Network-Specific Properties

Neural Network-Specific properties describe how the sparsity measure should behave when SNNs' parameters are adjusted or when the network is expanded.

1.Sparsity Changes with Weight Adjustment

Definition: For each spiking neuron $i$, there exists a $\beta_i > 0$ such that for any positive $\alpha$, adjusting the weight matrix $W$ by adding $\alpha$ to $w_i$ results in an increase in the sparsity measure.

Proof for $I(W)$: The sparsity measure $I(W)$ is sensitive to the concentration of nonzero weights. When a small weight adjustment is made that concentrates weights more in certain neurons, the sparsity decreases (because nonzero weights become more focused). This leads to an increase in $\|W\|_p$, and thus $I(W)$ will increase, as expected.

2. Adding Zero Weights Increases Sparsity

Definition: Adding zero weights to the network increases the sparsity measure, as the nonzero weights are now less concentrated.

Proof for $I(W)$: Adding zero weights results in more zero entries in the weight matrix for SNNs, decreasing the concentration of nonzero elements. This leads to a lower $\|W\|_p$, which, based on the formula, increases $I(W)$. Therefore, $I(W)$ correctly reflects the increase in sparsity when zero weights are added.

## B Reproducibility Checklist

This paper:

- Includes a conceptual outline and/or pseudocode description of AI methods introduced (yes)

- Clearly delineates statements that are opinions, hypothesis, and speculation from objective facts and results (yes)
- Provides well marked pedagogical references for less-familiare readers to gain background necessary to replicate the paper (yes)

Does this paper make theoretical contributions? (yes) If yes, please complete the list below.

- All assumptions and restrictions are stated clearly and formally. (yes)
- All novel claims are stated formally (e.g., in theorem statements). (yes)
- Proofs of all novel claims are included. (NA)
- Proof sketches or intuitions are given for complex and/or novel results. (NA)
- Appropriate citations to theoretical tools used are given. (yes)
- All theoretical claims are demonstrated empirically to hold. (yes)
- All experimental code used to eliminate or disprove claims is included. (yes)

Does this paper rely on one or more datasets? (yes)

If yes, please complete the list below.

- A motivation is given for why the experiments are conducted on the selected datasets (yes)
- All novel datasets introduced in this paper are included in a data appendix. (NA)
- All novel datasets introduced in this paper will be made publicly available upon publication of the paper with a license that allows free usage for research purposes. (NA)
- All datasets drawn from the existing literature (potentially including authors' own previously published work) are accompanied by appropriate citations. (yes)
- All datasets drawn from the existing literature (potentially including authors' own previously published work) are publicly available. (yes)
- All datasets that are not publicly available are described in detail, with explanation why publicly available alternatives are not scientifically satisficing. (yes)

Does this paper include computational experiments? (yes)

If yes, please complete the list below.

- Any code required for pre-processing data is included in the appendix. (yes).
- All source code required for conducting and analyzing the experiments is included in a code appendix. (yes)
- All source code required for conducting and analyzing the experiments will be made publicly available upon publication of the paper with a license that allows free usage for research purposes. (yes)
- All source code implementing new methods have comments detailing the implementation, with references to the paper where each step comes from (yes)
- If an algorithm depends on randomness, then the method used for setting seeds is described in a way sufficient to allow replication of results. (yes)
- This paper specifies the computing infrastructure used for running experiments (hardware and software), including GPU/CPU models; amount of memory; operating system; names and versions of relevant software libraries and frameworks. (yes)
- This paper formally describes evaluation metrics used and explains the motivation for choosing these metrics. (yes)
- This paper states the number of algorithm runs used to compute each reported result. (yes)
- Analysis of experiments goes beyond single-dimensional summaries of performance (e.g., average; median) to include measures of variation, confidence, or other distributional information. (yes)

- The significance of any improvement or decrease in performance is judged using appropriate statistical tests (e.g., Wilcoxon signed-rank). (yes)
- This paper lists all final (hyper-)parameters used for each model/algorithm in the paper's experiments. (yes)
- This paper states the number and range of values tried per (hyper-) parameter during development of the paper, along with the criterion used for selecting the final parameter setting. (yes)