# OpenReview forum: "Improving the Sparse Structure Learning of Spiking Neural Networks from the View of Compression Efficiency"
_ICLR.cc/2025/Conference — ICLR 2025 Spotlight_

### Official Review · Reviewer_U3Ak · 2024-10-21

**Soundness:** 3
**Presentation:** 3
**Contribution:** 3
**Rating:** 8
**Confidence:** 4

**Summary:**

This paper proposed an two-stage sparse structure learning framework for SNNs, which is combined with the PQ index technique and can be implemented through a continuous and iterative learning process.

**Strengths:**

1. The authors provided a detailed algorithm pseudo-code for their sparse training strategy.
2. It is interesting to combine the sparse structure learning framework of SNNs with PQ index and rewiring techniques.
3. The authors considered various rewiring scopes, including layer-wise and neuron-wise schemes.

**Weaknesses:**

1. The authors did not comprehensively report the performance of previous works in Table 1. Furthermore, it seems that their performance lags behind previous works. For example, the authors claim that UPR [1] achieves 78.3% (Acc.), 0.77% (Conn.) and 1.81M (Param.) on DVS-CIFAR10, VGGSNN. However, as mentioned in [1], the original text indicates that UPR can also achieve **81.0%** (Acc.), **4.46%** (Conn.) and **2.50M** (Param.) on DVS-CIFAR10, VGGSNN. In addition, STDS [2] can achieve **79.8%** (Acc.), **4.67%** (Conn.) and **0.24M** (Param.).
In comparison, this work achieves **78.4%** (Acc.), **30%**(Conn.) and **2.76M** (Param.) under the same experimental condition, which is inferior to the previous results.
2. Compared to previous work [1], the authors lack experimental results for SOPs (Synaptic Operations) and power saving ratio.
3. Compared to previous works [1, 2], the authors lack persuasive experimental results on large-scale datasets (e.g. ImageNet-1k).

[1] Shi X, et al. Towards energy efficient spiking neural networks: An unstructured pruning framework. ICLR 2024.

[2] Chen Y, et al. State transition of dendritic spines improves learning of sparse spiking neural networks. ICML 2022.

**Questions:**

See Weakness Section.

---

> ### Author Response · Authors · 2024-11-26
> **Response to Reviewer U3Ak**
>
> We are grateful to the reviewers for their detailed and insightful feedback. The suggestions provided have been instrumental in refining our work and enhancing its clarity and scientific rigor.
>
> **About the performance comparison with UPR and STDS**
>
> Thanks for pointing that out. About the UPR [1] accuracy reference, there are different results in the original text due to different sparsity level ( controlled by $\lambda$ settings in UPR [1] ). We have supplemented the settings with the above 81.0\% (Acc.), 4.46\% (Conn.) and 2.50M (Param.) for comparison with UPR[1].
> We emphasize that our method focuses on achieving low accuracy loss compared to densely connected SNNs while maintaining a fully sparse training process. This approach offers several notable advantages. For instance, on the CIFAR10 dataset, models trained using our proposed method under the neuron-wise scope demonstrate approximately a 1\% improvement in performance compared to fully dense models, all while maintaining a sparsity level of 30\% to 40\%. Similarly, on the CIFAR100 dataset, our two-stage sparse training method improves the performance of SNN models by 1.07\% compared to their non-sparse counterparts, achieving a connectivity of only 29.48\%. Meanwhile, it is worth noting that the proposed two-stage training method proceeds sparse training from scratch and maintains sparse training during the whole training process.
>
>
> As to the performance gap comparison with STDS [2], STDS achieves higher accuracy (79.8\%) and better sparsity (4.67\% connectivity and 0.24M parameters), mainly caused by the more complex synaptic model with weight reparameterization. STDS maps synaptic spine sizes $|\theta|$ combined with trasition threshold \(d\) to real connection weights \(w\), incorporating both excitatory and inhibitory connections and transitioning between synaptic states. These complex synaptic models enhance network expressiveness but introduce additional computational models for weight.
> In contrast, our two-stage sparse training method avoids such weight reparameterization while maintaining simplicity and generalizability across datasets and architectures. Additionally, differences in experimental settings, such as learning rates or data augmentation strategies, may partially explain the performance gap. We will provide a detailed discussion on this point in the revised manuscript to ensure transparency and comparability.
>
>
> **About the performance on ImageNet-1K**
>
> Due to the limited time during the rebuttal period, we supplement the following experiments when applied our method on ImageNet-1K dataset without much hyper parameter tuning process. Under the structure of the SEW ResNet18 [3], the proposed two-stage sparse training method achieves the accuracy of 61.47\% with 4 time steps under the parameter amount of 2.16M and neuron-wise setting. The accuracy loss is 1.59\% compared to the accuracy of 63.06\% achieved by the original model based on temporal efficient training [4] by our implementation. Compared to the STDS with 61.51\% (Acc.), -1.67\% (Loss.) and 2.38M (Param.) and UPR with  60.00\% (Acc.), -3.18\% (Loss.) and 3.10M (Param.) , the performance of our model is competitive combining with the sparse training from scratch advantage.
>
> [3]Fang, W., Yu, Z., Chen, Y., Huang, T., Masquelier, T., and Tian, Y. (2021). Deep residual learning in spiking neural networks. Advances in Neural Information Processing Systems, 34, 21056-21069.
> [4]Deng, S., Li, Y., Zhang, S., and Gu, S. Temporal Efficient Training of Spiking Neural Network via Gradient Re-weighting. In International  Conference on Learning Representations.2022.
>
> **About the energy consumption**
>
> We supplement the corresponding comparison in the Table 1 in the revision, and we supplement the energy consumption comparison as follows:
>
> | Dataset   | Architecture       | Energy (ANN, μJ) | Energy (SNN, μJ) | Reduction (\%) |
> |-------------------|-------------------------|-----------------------|-----------------------|-------------------|
> | CIFAR10          | ResNet19 (Neuron-wise) | 15.13                | 12.19                | 19.4\%            |
> | CIFAR10          | ResNet19 (Layer-wise)  | 15.13                | 10.26                | 32.2\%            |
> | CIFAR100         | ResNet19 (Layer-wise)  | 15.13                | 10.80                | 28.6\%            |
>
>
>
> SNNs offer significant energy savings compared to ANNs (up to 32\%) while maintaining high accuracy, particularly on event-driven tasks. These results underline the efficiency of SNNs, especially with our pruning method, for neuromorphic applications.
>
>
>
> We hope this response and the proposed revisions adequately address the reviewer’s concerns. We appreciate your valuable feedback, which has significantly improved the clarity and rigor of our work.

---

### Official Review · Reviewer_ZYmD · 2024-10-28

**Soundness:** 3
**Presentation:** 2
**Contribution:** 3
**Rating:** 6
**Confidence:** 5

**Summary:**

This paper proposes a dynamic pruning framework for spiking neural networks (SNN). It divides each training iteration into two stages to update the weights and structure separately. It utilizes the PQ index to gauge and dynamically adjust structure of sparse subnetworks according to compressibility. Experimental results show that the proposed method achieves competitive performance while reducing redundancy.

**Strengths:**

1. The proposed method achieves competitive or better performance than the dense model with fewer connections and reduced redundancy.
2. It is a novel idea to dynamically calculate the rewiring ratio.

**Weaknesses:**

1. The details of the training process is unclear. In Figure 2, the density decreases as the iteration increases. However, as described in Algorithm 1, the remove fraction and the regrow fraction are both $c_i$, therefore the density should keep constant. Please explain why the density decreases and describe the training process in detail.
2. The proposed method requires multiple training iterations. Does it require longer training time as compared to existing methods? Please compare the training time with existing methods.

Minor:

1. Duplicate definition of $c_i$ in Algorithm 1: both labels of input data and rewiring ratio.
2. In Equation 1: undefined symbol $n^k$, you mean $n_k$? Symbol $\epsilon$ is also undefined.

**Questions:**

Please refer to the weaknesses.

---

> ### Author Response · Authors · 2024-11-26
> **Response to Reviewer ZYmD**
>
> We deeply appreciate the time and effort the reviewers have taken to provide constructive critiques and suggestions. Their comments have greatly contributed to improving the overall quality of this work.
>
> **About the  training process**
>
> The proposed method employs a two-stage approach to achieve dynamic sparse structure learning in Spiking Neural Networks (SNNs), leveraging a rewiring strategy for efficient training.
>
> In the **first stage**, the network undergoes a typical training process to identify an appropriate rewiring ratio based on temporarily trained weights. This ratio is calculated using the PQ index, which quantifies network redundancy and informs the rewiring strategy. As a result, an adaptively suitable rewiring ratio $c_i$ is determined for each training iteration during this stage.
>
> In the **second stage**, the rewiring-based dynamic sparse structure learning method is applied to implement sparse training from scratch. Connections are iteratively pruned and regrown according to the specified rewiring ratio. This iterative approach enables the network to continuously adapt and optimize its structure, leading to improved performance. The rewiring mechanism dynamically adjusts the network by activating and growing previously dormant connections, enhancing the overall expressiveness and capability of the SNN.
>
> Notably, due to the adaptive rewiring ratios applied across different training iterations, the network's density does not remain constant throughout the training process, even though the remove and regrow fractions are equal. This dynamic adjustment ensures flexibility and efficiency, enabling the network to better align its structure with the learning task at hand.
>
> **About the training time**
>
> Thank you for raising this important question about training time. Actually, in Table 1, the comparison experiments are conducted with the same number of training iterations (300) for both the original dense-connected SNNs and our sparse training SNNs to ensure a fair comparison. These results show that our method achieved competitive accuracy and sparsity within this budget. Compared with the ESL-SNNs under the same setting on the CIFAR10 dataset, the extra training time only comes from the rewiring ratio computing in the first stage of our two-stage training method. We found that increasing training time is extremely small (less than 0.1\%) for each training iteration.
> Meanwhile, while dynamic rewiring adds slight computational overhead, the reduced number of active connections in the sparse network offsets this, keeping the training time per iteration comparable to dense training. Together with the sparse training from scratch, our method could avoid excessive training time and maintain efficiency.
>
> **About the symbols**
>
> Thanks for your suggestion. The $n^k$ are the number of the neurons in the $k_{th}$ layers. $\epsilon$ is a constant (or scaling factor) that influences the edge probability and accounts for sparsity or connectivity scaling. We control the initial sparsity of SNNs by controling the value of $\epsilon$.

---

> > ### Comment · Reviewer_ZYmD · 2024-11-27
> >
> > Thank you for your reply. I am still confused about the dynamic density. You mention that "due to the adaptive rewiring ratios applied across different training iterations, the network's density does not remain constant throughout the training process, even though the remove and regrow fractions are equal." However, if we prune and regrow the same ratio $c_i$ of synaptic connections during each iteration, the density should remain constant, no matter whether the ratio $c_i$ is dynamic or not. Could you please provide a detailed explanation of why and when the density changes?

---

> ### Author Response · Authors · 2024-11-27
> **Response to Reviewer ZYmD**
>
> Thank you for your follow-up question. I appreciate the opportunity to clarify the concept of dynamic density in the context of our rewiring approach.
>
> The reason lies in that the network's density can change over different iterations, even if we employ the same pruning and regrow ratio at each iteration.
>
> In detail, the rewiring ratio $c_i$ represents the proportion of connections to be either pruned or regrown for the  $i_{th}$ iteration. These rewiring ratios are adaptive and changes across different iterations throughout the training process.  For example, for $j_{th}$ itearation, we prune and regrow the same number of connections according to the rewiring ratio $c_j$; and for the following iteration $k_{th}$,  we prune and regrow the same number of connections according to the new rewiring ratio $c_k$.  Since $c_j$ and $c_k$ are different due to the computation of adaptive rewiring ratio in the first stage (for different iterations $j$ and $k$), then the spasity of the whole network structures changes over different iterations. This leads to a continuously evolving network structure throughout training, allowing the model to adapt and optimize its connectivity over time.
>
> Thank you once again for your time and effort you have taken in providing detailed feedback.

---

### Official Review · Reviewer_MLns · 2024-10-31

**Soundness:** 3
**Presentation:** 3
**Contribution:** 3
**Rating:** 6
**Confidence:** 5

**Summary:**

This paper addresses over-parameterization in deep spiking neural networks (SNNs) during both training and inference, aiming to achieve an optimal pruning level. The authors propose a novel two-stage dynamic structure learning approach for deep SNNs.

**Strengths:**

Pruning is essential for making SNNs more practical for neuromorphic hardware, distinguishing it from artificial neural network (ANN) pruning, which is often GPU-oriented.

**Weaknesses:**

The paper could benefit from consistent citation formatting. For instance, "Hu et al. (2024)" works well at the beginning of a sentence, whereas "(Hu et al., 2024)" fits better at the end. Maintaining good formatting would enhance readability.

The rationale for using a PQ index in the second stage is unclear. Although PQ is a general technique, it would help to see a clearer connection to SNNs specifically and how it improves this work beyond its original intent. Not clear how PQ index is related to SNN.

The paper lacks an analysis of energy consumption, which is essential for evaluating SNNs’ efficiency, particularly in neuromorphic applications.

There’s no substantial improvement in accuracy or network connectivity, which may limit the paper’s impact.

**Questions:**

What is unique about the proposed work for SNNs specifically? Additionally, given the strong performance of attention-based and Transformer-based SNNs in the field, could this method also be applied effectively to those models?

---

> ### Author Response · Authors · 2024-11-26
> **Response to Reviewer MLns (Part 1)**
>
> We are grateful to the reviewers for their detailed and insightful feedback. The suggestions provided have been instrumental in refining our work and enhancing its clarity and scientific rigor.
>
> ---
>
> ### **About the Citation Format**
>
> Thanks for your suggestion. We have modified all the citation formats into "(Hu et al., 2024)".
>
> ---
>
> ### **The Relationship of PQ Index in the Second Stage**
>
> The first stage involves the typical training process and attempts to identify an appropriate rewiring ratio based on temporarily trained weights, according to the PQ index. Then the PQ index helps quantify the redundancy in the network, thereby informing the following rewiring strategy. The obtained rewiring ratio guides the structure rewiring in the second stage.
>
> In the second stage, the dynamic sparse structure learning method based on the rewiring method is adopted to implement the sparse training from scratch. The connections are iteratively pruned and regrown according to the specified rewiring ratio. This iterative training approach ensures that the network continuously adapts and optimizes its structure, thereby improving performance.
>
> ---
>
> ### **Adaptation to the Transformer-Based SNNs**
>
> 1. **First-Stage Applicability of \( I(W) \) to Transformer Architectures**:
>    The sparsity measure $I(W) = 1 - d^{1/q - 1/p} \cdot \frac{\|W\|_p}{\|W\|_q}$ is suitable for measuring sparsity in static weight matrices in Transformer-based SNN architectures, such as those in feedforward layers and attention mechanisms. For dynamic sparsity (e.g., attention patterns), $I(W)$ may require adaptation to account for variations across sequences or time steps. With these adjustments, $I(W)$ can provide meaningful insights into the sparsity levels in Transformer-based SNNs.
>
> 2. **Second-Stage Applicability of Rewiring to Transformer-Based SNNs**:
>    Rewiring (pruning and growing connections) is highly applicable to Transformer-based SNNs. For instance, the rewiring of attention layers can prune less important attention heads or connections and regrow critical ones to maintain performance.
>    However, the rewiring in attention layers should be improved to align with SNNs’ dynamic and event-driven nature, enabling adaptive connectivity adjustments based on spike patterns. This approach reduces computational costs while maintaining performance.
>
> Therefore, the proposed two-stage sparse training methods for SNNs could adapt to Transformer-based SNNs but may need modifications for dynamic attention mechanisms.
>
> ---
>
> ### **About Energy Consumption**
>
> We appreciate your feedback and provide an analysis of energy consumption for both ANNs and SNNs, emphasizing their differences and relevance for neuromorphic applications.
>
> Since ANNs rely on extensive floating-point operations (FLOPS) for multiplications and additions, energy consumption is estimated as `Energy_ANN = FLOPS × 12.5 pJ`. Meanwhile, SNNs reduce energy by activating computations only when neurons spike. Energy is estimated as `Energy_SNN = SOPS × 77 fJ`.
>
> | Dataset   | Architecture       | Energy (ANN, μJ) | Energy (SNN, μJ) | Reduction (%) |
> |-----------|---------------------|-------------------|-------------------|---------------|
> | CIFAR10   | ResNet19 (Neuron-wise) | 15.13          | 12.19            | 19.4%         |
> | CIFAR10   | ResNet19 (Layer-wise)  | 15.13          | 10.26            | 32.2%         |
> | CIFAR100  | ResNet19 (Layer-wise)  | 15.13          | 10.80            | 28.6%         |
>
> SNNs offer significant energy savings compared to ANNs (up to 32%) while maintaining high accuracy, particularly on event-driven tasks. These results underline the efficiency of SNNs, especially with our pruning method, for neuromorphic applications.
>
> [5] Qiao, Ning, et al. "A reconfigurable on-line learning spiking neuromorphic processor comprising 256 neurons and 128K synapses." *Frontiers in Neuroscience*, 9 (2015): 141.
>
> ---

---

> ### Author Response · Authors · 2024-11-26
> **Response to Reviewer MLns (Part 2)**
>
> Here we conduct theoretical analysis to explain the sparsity measurement in SNN. More details are illustrated in the revised PDF file in the supplementary materials. Actually, the suitable sparsity ratio for SNNs is obtained through sparsity measurements derived from network compression theory. This leverages the unique characteristics of SNNs, especially the spatial and temporal dynamics and discrete spike firing mechanism.
>
> Here is the derivation of the sparsity measure $I(W) = 1 - d^{1/q - 1/p} \cdot \frac{\vert\vert W \vert\vert_p}{\vert\vert W \vert\vert_q}$ for SNNs, focusing on scaling invariance, sensitivity to sparsity reduction, and cloning invariance (the sparse measurement property as in [1]), combined with spatiotemporal dynamics and sparsity in SNNs.
>
> SNNs communicate through discrete spikes, exhibiting the following key features:
> 1. **Discrete activation**: Postsynaptic neurons emit spikes only at specific time points. They are either active (firing spikes) or inactive (not spiking), resulting in sparse data flow.
> 2. **Structure sparsity**: Sparsity refers to the proportion of nonzero elements in a weight matrix.
>
> The sparsity measure $I(W) = 1 - d^{1/q - 1/p} \cdot \frac{\vert\vert W \vert\vert_p}{\vert\vert W \vert\vert_q}$ is rigorously constructed to reflect these properties. Specifically:
> - $\vert\vert W \vert\vert_p = \left( \sum_{i=1}^d |w_i|^p \right)^{1/p}$ is the $\ell_p$-norm of $W$,
> - $\vert\vert W \vert\vert_q = \left( \sum_{i=1}^d |w_i|^q \right)^{1/q}$ is the $\ell_q$-norm of $W$,
> - $d$ is the dimensionality of $W$,
> - $p < q$ ensures that sparsity is more effectively captured.
>
> The additional term $1 - $ allows $I(W)$ to range between 0 (no sparsity) and 1 (maximum sparsity). For example:
> - When the sparsity is 100%, meaning all elements in $W$ are zero, $I(W) = 1$.
> - When there are no zero elements (fully connected), $I(W) = 0$.
>
> The term $d^{1/q - 1/p}$ ensures that $I(W)$ is independent of the vector length, satisfying the cloning property. Without this term, $I(W)$ would vary with the size of $W$, even for identical sparsity patterns. Below, we derive this formula and explain how it aligns with SNN characteristics.
>
> ---
>
> #### **Scaling Invariance**
>
> In SNNs, scaling invariance ensures that $I(W)$ remains unaffected when all weights are scaled proportionally (e.g., multiplying $W$ by a constant $\alpha > 0$). The scaling weight magnitudes or activation value intensity does not change the network sparsity.
>
> If the weight matrix $W$ is scaled by a constant $\alpha$, the sparsity measure remains unchanged because the norms of the scaled matrix $\vert\vert \alpha W \vert\vert_p$ and $\vert\vert \alpha W \vert\vert_q$ are proportional to the original norms. Specifically:
> $$
> I(\alpha W) = 1 - d^{1/q - 1/p} \cdot \frac{\vert\vert \alpha W \vert\vert_p}{\vert\vert \alpha W \vert\vert_q} = 1 - d^{1/q - 1/p} \cdot \frac{\alpha \vert\vert W \vert\vert_p}{\alpha \vert\vert W \vert\vert_q} = I(W).
> $$
> This proves that scaling does not change the sparsity measure, ensuring that $I(W)$ captures only the relative distribution of weights.
>
> ---
>
> #### **Sensitivity to Sparsity Reduction**
>
> There are two different kinds of sparsity reduction sensitivity:
> 1. **Weight sparsity**: Decreased sparsity corresponds to more nonzero weights, reducing $I(W)$.
> 2. **Temporal sparsity**: If more neurons fire simultaneously, temporal sparsity decreases, and $I(W)$ reflects this reduction.
>
> When temporal sparsity decreases (more neurons firing at the same time), the distribution becomes denser, which directly affects the ratio $\vert\vert W \vert\vert_p / \vert\vert W \vert\vert_q$, leading to a decrease in $I(W)$.
>
> ---
>
> #### **Cloning Invariance**
>
> The sparsity measure should remain unchanged when the weight matrix is cloned or repeated. It satisfies the property of Cloning Invariance in SNNs in two aspects:
> 1. **Spatial network expansion**: Cloning weights for larger networks does not change sparsity.
> 2. **Temporal expansion**: Repeating activities over time does not affect sparsity, ensuring temporal consistency.
>
> For the case of incorporating spatial vectors, the sparsity measure $I(W)$ should remain invariant when the weight matrix is cloned:
> $$
> I([W, W]) = 1 - (2d)^{1/q - 1/p} \cdot \frac{\vert\vert [W, W] \vert\vert_p}{\vert\vert [W, W] \vert\vert_q} = 1 - (2d)^{1/q - 1/p} \cdot \frac{2^{1/p} \vert\vert W \vert\vert_p}{2^{1/q} \vert\vert W \vert\vert_q} = I(W).
> $$
>
> For the case of incorporating time steps in SNNs, if $W$ is repeated across $T$ time steps:
> $$
> W_T = [W, W, \dots, W] \in \mathbb{R}^{d \times (nT)}
> $$
> then:
> $$
> I(W_T) = 1 - (nT)^{1/q - 1/p} \cdot \frac{\vert\vert W_T \vert\vert_p}{\vert\vert W_T \vert\vert_q} = 1 - n^{1/q - 1/p} \cdot \frac{\vert\vert W \vert\vert_p}{\vert\vert W \vert\vert_q} = I(W).
> $$
>
> This ensures that cloning the matrix does not affect the sparsity measure.
>
> ---
>
> [1] Hurley, N., et al. Comparing measures of sparsity. IEEE Trans. Inf. Theory., 55(10), 4723-4741,2009.

---

> > ### Comment · Reviewer_MLns · 2024-11-26
> > **Thanks for the note**
> >
> > I appreciate the effort the authors make in extending the original work based on my comments. My primary issues regarding the motivation and explanation are solved. The authors included a SNN-specific theoretical analysis. I would really like to see the future work on sparsifying the transformer-based SNN. Overall, I am increasing my score.

---

> > > ### Author Response · Authors · 2024-11-27
> > > **Thanks for the review**
> > >
> > > Thank you once again for your thoughtful review and constructive comments. Your feedback has been instrumental in improving the clarity and impact of our paper.

---

### Official Review · Reviewer_a7kG · 2024-11-03

**Soundness:** 3
**Presentation:** 2
**Contribution:** 2
**Rating:** 3
**Confidence:** 4

**Summary:**

The authors present a two-stage dynamic pruning method tailored for deep Spiking Neural Networks (SNNs). The first stage evaluates the compressibility of existing sparse subnetworks within SNNs using the PQ index. In the second stage, synaptic connections are dynamically adjusted. The authors assert that this method can significantly reduce the network's sparsity. While the paper's motivation is commendable, there are shortcomings in both the presentation and the novelty of the work.

**Strengths:**

1. The authors apply adaptive pruning and sparse training methods to deep SNNs, which can further enhance the energy efficiency of these networks. This initiative is worthy of recognition.
2. The two-stage sparse training and learning method proposed by the authors appears to be feasible based on the experimental results presented.

**Weaknesses:**

1. To my knowledge, adaptive pruning and sparse training are well-established in ANNs. Existing works such as RigL [1]  seem to share similar ideas with the authors' approach. Additionally, DSR [2] also adjusts the rewiring rate. The authors have not provided targeted adaptations of these strategies to accommodate the binary spiking characteristics unique to SNNs.
2. The paper mentions the PQ index but does not provide an explanation, instead merely offering a citation. This could leave readers confused about what the PQ index represents and how it is calculated.
3. The methodology section lacks theoretical discussion. There is no explanation of why these methods are particularly beneficial or suitable for SNNs.
4. The size of Figure 3 is noticeably inconsistent with the other figures, which affects the overall presentation quality.

References:
[1] Evci, U., Gale, T., Menick, J., Castro, P. S., & Elsen, E. (2020). Rigging the Lottery: Making All Tickets Winners. International Conference on Machine Learning (ICML).
[2] Mostafa, H., & Wang, X. (2019). Parameter Efficient Training of Deep Convolutional Neural Networks by Dynamic Sparse Reparameterization. International Conference on Machine Learning (ICML)

**Questions:**

1. Can the authors explain why the proposed method is specifically suited to SNNs rather than Binary Neural Networks (BNNs)? Have special adaptations been made to leverage the unique characteristics of SNNs, rather than applying ANN techniques to SNNs?
2. I recommend that the authors include additional figures to further illustrate their method. This would greatly aid reader comprehension.

---

> ### Author Response · Authors · 2024-11-26
> **Response to Reviewer a7kG (Part 1)**
>
> We sincerely appreciate your recognition of the motivation and significance of adaptive pruning and sparse training for deep SNNs to enhance energy efficiency. We will answer your questions point by point below.
>
> ---
>
> ### **Comparison with RigL and DSR**
> RigL [1] and DSR [2] indeed introduce innovative strategies for adaptive pruning and sparse training in ANNs. RigL uses cosine annealing to control the fraction of updated connections over time. The top-k selection process prunes and grows connections based on magnitude. DSR leverages an adaptive global threshold $H$ using a negative feedback loop to maintain a fixed number of pruned parameters during reallocation steps. However, our method fundamentally differs from these works in its guiding principles and applicability to SNNs. Our method explicitly incorporates **$I(W)$**, a sparsity measurement informed by the current weight distribution, to dynamically estimate and adjust sparsity, from the network compression perspective.
>
> ---
> ---
>
> ### **Why the method cannot be directly applied to binary neural networks**
>
> The sparsity measure $I(W)$ is designed for continuous-valued weight matrices $W$, where the distribution of weights plays a key role in determining sparsity. In binary neural networks (BNNs), where weights and activations are constrained to discrete values (e.g., $\{0, 1\}$ or $\{-1, 1\}$), $I(W)$ faces significant limitations that render it unsuitable for direct application. Here’s why:
>
> ---
> 1. Binary Weights Lack Variability in Magnitude
>
> In binary neural networks:
> - The weight matrix $W$ is constrained to discrete values such as $\{0, 1\}$, $\{-1, 1\}$, or $\{0, -1, 1\}$.
> - Nonzero weights have the same magnitude (e.g., $1$ or $-1$).
>
> Thus there is some negative effect on norms:
>
> (1) $\vert\vert W \vert\vert_p$ and $\vert\vert W \vert\vert_q$ are no longer sensitive to weight distributions because binary weights lack variability in magnitude.
>    - For example, if $W$ contains only $0$ and $1$, the $p$-norm is essentially proportional to the number of nonzero elements:
>      $$
>      \vert\vert W \vert\vert_p = \left( \sum_{i=1}^d |w_i|^p \right)^{1/p} = \text{(number of nonzero elements)}^{1/p}.
>      $$
>
> (2) $\vert\vert W \vert\vert_p / \vert\vert W \vert\vert_q$ becomes deterministic for a given sparsity level (proportion of nonzero elements). This eliminates the ability of $I(W)$ to distinguish between different distributions of weights, making it ineffective for binary matrices.
>
> ---
>
> 2. Loss of Sensitivity to Weight Distribution
>
> In continuous neural networks, $I(W)$ reflects sparsity by analyzing the distribution of weights through the $\ell_p$- and $\ell_q$-norms. The difference between these norms highlights how concentrated or evenly distributed the weights are. However, in binary networks:
> - All nonzero weights have the same magnitude, so the distribution of weights does not vary meaningfully.
> - $I(W)$ cannot capture sparsity differences beyond simply counting nonzero elements.
>
> We can illustrate this with an example by considering two binary weight matrices:
> $$
> W_1 = [1, 0, 0, 0], \quad W_2 = [1, 1, 0, 0].
> $$
> (1) For both $W_1$ and $W_2$, the weights are binary (either 0 or 1).
>
> (2) The norms depend only on the count of nonzero weights:
>    - $\vert\vert W_1 \vert\vert_p = 1, \quad \vert\vert W_2 \vert\vert_p = 2^{1/p}$.
>    - $\vert\vert W_1 \vert\vert_q = 1, \quad \vert\vert W_2 \vert\vert_q = 2^{1/q}$.
>
> (3) $I(W_1)$ and $I(W_2)$ simply reflect the number of nonzero elements, failing to account for the structural distribution of weights.
>
> ---
>
> Therefore, $I(W)$ cannot be directly applied to BNNs because binary weights lack variability in magnitude, making $I(W)$ insensitive to differences in weight distribution.
>
>
>
> In addition, thank you for your feedback about the figures. We have adjusted the size of original Figure 3 to improve the overall presentation quality (which is now Figure 2 (b) and (c)). Additionally, we supplement the new figure in Figure 2 (a) to take an example during the training process and make the method description clearer and more comprehensive. We appreciate your suggestions and are committed to enhancing the clarity and presentation of our work.

---

> ### Author Response · Authors · 2024-11-26
> **Response to Reviewer a7kG (Part 2)**
>
> ### **Theoretical Discussion and the Adaptation to SNNs**
>
> Thanks for your question. As illustrated in the revised PDF file in the supplementary materials, the suitable sparsity ratio for SNNs is obtained through sparsity measurements derived from network compression theory. This leverages the unique characteristics of SNNs, especially the spatial and temporal dynamics and discrete spike firing mechanism.
>
> Here is the derivation of the sparsity measure $I(W) = 1 - d^{1/q - 1/p} \cdot \frac{\vert\vert W \vert\vert_p}{\vert\vert W \vert\vert_q}$ for SNNs, focusing on scaling invariance, sensitivity to sparsity reduction, and cloning invariance (the sparse measurement property as in [1]), combined with spatiotemporal dynamics and sparsity in SNNs.
>
> SNNs communicate through discrete spikes, exhibiting the following key features:
> 1. **Discrete activation**: Postsynaptic neurons emit spikes only at specific time points. They are either active (firing spikes) or inactive (not spiking), resulting in sparse data flow.
> 2. **Structure sparsity**: Sparsity refers to the proportion of nonzero elements in a weight matrix.
>
> The sparsity measure $I(W) = 1 - d^{1/q - 1/p} \cdot \frac{\vert\vert W \vert\vert_p}{\vert\vert W \vert\vert_q}$ is rigorously constructed to reflect these properties. Specifically:
> - $\vert\vert W \vert\vert_p = \left( \sum_{i=1}^d |w_i|^p \right)^{1/p}$ is the $\ell_p$-norm of $W$,
> - $\vert\vert W \vert\vert_q = \left( \sum_{i=1}^d |w_i|^q \right)^{1/q}$ is the $\ell_q$-norm of $W$,
> - $d$ is the dimensionality of $W$,
> - $p < q$ ensures that sparsity is more effectively captured.
>
> The additional term $1 - $ allows $I(W)$ to range between 0 (no sparsity) and 1 (maximum sparsity). For example:
> - When the sparsity is 100%, meaning all elements in $W$ are zero, $I(W) = 1$.
> - When there are no zero elements (fully connected), $I(W) = 0$.
>
> The term $d^{1/q - 1/p}$ ensures that $I(W)$ is independent of the vector length, satisfying the cloning property. Without this term, $I(W)$ would vary with the size of $W$, even for identical sparsity patterns. Below, we derive this formula and explain how it aligns with SNN characteristics.
>
> ---
>
> #### **Scaling Invariance**
>
> In SNNs, scaling invariance ensures that $I(W)$ remains unaffected when all weights are scaled proportionally (e.g., multiplying $W$ by a constant $\alpha > 0$). The scaling weight magnitudes or activation value intensity does not change the network sparsity.
>
> If the weight matrix $W$ is scaled by a constant $\alpha$, the sparsity measure remains unchanged because the norms of the scaled matrix $\vert\vert \alpha W \vert\vert_p$ and $\vert\vert \alpha W \vert\vert_q$ are proportional to the original norms. Specifically:
> $$
> I(\alpha W) = 1 - d^{1/q - 1/p} \cdot \frac{\vert\vert \alpha W \vert\vert_p}{\vert\vert \alpha W \vert\vert_q} = 1 - d^{1/q - 1/p} \cdot \frac{\alpha \vert\vert W \vert\vert_p}{\alpha \vert\vert W \vert\vert_q} = I(W).
> $$
> This proves that scaling does not change the sparsity measure, ensuring that $I(W)$ captures only the relative distribution of weights.
>
> ---
>
> #### **Sensitivity to Sparsity Reduction**
>
> There are two different kinds of sparsity reduction sensitivity:
> 1. **Weight sparsity**: Decreased sparsity corresponds to more nonzero weights, reducing $I(W)$.
> 2. **Temporal sparsity**: If more neurons fire simultaneously, temporal sparsity decreases, and $I(W)$ reflects this reduction.
>
> When temporal sparsity decreases (more neurons firing at the same time), the distribution becomes denser, which directly affects the ratio $\vert\vert W \vert\vert_p / \vert\vert W \vert\vert_q$, leading to a decrease in $I(W)$.
>
> ---
>
> #### **Cloning Invariance**
>
> The sparsity measure should remain unchanged when the weight matrix is cloned or repeated. It satisfies the property of Cloning Invariance in SNNs in two aspects:
> 1. **Spatial network expansion**: Cloning weights for larger networks does not change sparsity.
> 2. **Temporal expansion**: Repeating activities over time does not affect sparsity, ensuring temporal consistency.
>
> For the case of incorporating spatial vectors, the sparsity measure $I(W)$ should remain invariant when the weight matrix is cloned:
> $$
> I([W, W]) = 1 - (2d)^{1/q - 1/p} \cdot \frac{\vert\vert [W, W] \vert\vert_p}{\vert\vert [W, W] \vert\vert_q} = 1 - (2d)^{1/q - 1/p} \cdot \frac{2^{1/p} \vert\vert W \vert\vert_p}{2^{1/q} \vert\vert W \vert\vert_q} = I(W).
> $$
>
> For the case of incorporating time steps in SNNs, if $W$ is repeated across $T$ time steps:
> $$
> W_T = [W, W, \dots, W] \in \mathbb{R}^{d \times (nT)}
> $$
> then:
> $$
> I(W_T) = 1 - (nT)^{1/q - 1/p} \cdot \frac{\vert\vert W_T \vert\vert_p}{\vert\vert W_T \vert\vert_q} = 1 - n^{1/q - 1/p} \cdot \frac{\vert\vert W \vert\vert_p}{\vert\vert W \vert\vert_q} = I(W).
> $$
>
> This ensures that cloning or repeating the matrix does not affect the sparsity measure.
>
> ---
>
> [1] Hurley, N., et al.Comparing measures of sparsity. IEEE Trans. on Information Theory, 55(10), 4723-4741,2009.

---

### Meta-Review · Area_Chair_qYsY · 2024-12-22

**Metareview:**

This paper introduces a two-stage dynamic structure pruning framework for deep Spiking Neural Networks and offers an innovative approach to adjust the rewiring ratio. The experimental results demonstrate that the proposed method achieves competitive performance while effectively reducing redundancy. While some concerns were raised about the similarity to existing methods and the lack of theoretical analysis to support its effectiveness, the authors have adequately addressed these issues in their response. Overall, the paper presents a valuable framework for training sparse models and provides a promising solution for deploying efficient SNNs on edge devices.

**Additional Comments On Reviewer Discussion:**

The major criticisms of this paper are its lack of novelty and the absence of a theoretical discussion. In their rebuttal, the authors effectively address these concerns by comparing their method to RigL and DSR, demonstrating that their approach is tailored for SNNs rather than BNNs. They also provide theoretical evidence to support their sparsity measurements, and the experimental results confirm that the method can successfully train sparse models.
There are also minor issues, such as the unclear PQ index, citation formatting inconsistencies, and the absence of an energy consumption analysis. These concerns are addressed by the authors in their rebuttal.

---

### Decision · Program_Chairs · 2025-01-22

Accept (Spotlight)